# The Relationship Between Non-Traumatic Fat Embolism and Fat Embolism Syndrome (FES) in Patients with Cancer

**DOI:** 10.3390/diseases13060174

**Published:** 2025-05-30

**Authors:** Beáta Ágnes Borsay, Barbara Dóra Halasi, Zoltán Hendrik, Róbert Kristóf Pórszász, Katalin Károlyi, Teodóra Tóth, Péter Attila Gergely

**Affiliations:** 1Institute of Forensic Medicine, Faculty of Medicine, University of Debrecen, Nagyerdei krt. 98, H-4032 Debrecen, Hungary; halasi.barbara@med.unideb.hu (B.D.H.); hendrik.zoltan@med.unideb.hu (Z.H.); gergely.peter@med.unideb.hu (P.A.G.); 2Department of Pharmacology and Pharmacotherapy, Faculty of Medicine, University of Debrecen, Nagyerdei krt. 98, H-4032 Debrecen, Hungary; porszasz.robert@med.unideb.hu; 3Department of Pathology and Laboratory Medicine, Jefferson Einstein Hospital, 5501 Old York Rd, Philadelphia, PA 19141, USA; katalinkarolyi0@gmail.com; 4Department of Pathology, Clinical Center, Kenézy Gyula Campus, University of Debrecen, Bartók Béla út 2-26, H-4031 Debrecen, Hungary; teodoradrtoth@gmail.com

**Keywords:** pulmonary fat embolism, fat embolism syndrome, fat embolism in cancer, C-reactive protein level increasing, lipid staining, Oil Red O, fat globule

## Abstract

Background: Fat embolism and fat embolism syndrome are rare but well-known consequences of long bone fractures and orthopedic surgeries. These sources support the mechanical theory of their development. On the other hand, as an alternative pathway suggested by the biochemical theory, lipase activation and fat breakdown are also a possible background for lipid droplets appearing in the vasculature. According to Hulman’s theory, elevated C-reactive protein levels can facilitate calcium-dependent agglutination of very low-density proteins and chylomicrons forming fat globules. The level of this acute-phase protein can increase mainly in advanced-stage cancers but also has predictive or indicative value in treatment success. Methods: This study focused on strictly selected patients with different histological types and origins of cancer, as well as advanced cancer in approximately 90% of the deceased. After collecting the tissue samples, the frozen sections were stained with Oil Red O to detect fat emboli. Results: Less than 50% of the cases showed punctiform, non-clinically relevant pulmonary fat embolism, and fat embolism syndrome was identified in none of the cases. In one, non-advanced cancer case, punctiform kidney fat embolism was observed. Conclusions: The end-of-life anergic state of patients may influence the procedure. In the case of osseous metastases, since the intramedullary sinuses are affected, both the mechanical and the biochemical backgrounds may prevail and mediate fat embolism formation.

## 1. Introduction

Traumatic fat embolism and fat embolism syndrome are well-known conditions in traumatology, orthopedics, and forensic pathology practice. Soft tissue damage (mechanical or heat) and mainly long or sometimes pelvic bone fractures or orthopedic surgery (e.g., arthroplasty) are common causes of the formation of fat embolism [1,2]. In addition to those mentioned above, there are other non-traumatic reasons, including corticosteroid therapy, osteomyelitis, parenteral lipid infusion, and sickle cell disease, underlining its heterogeneous origins (Table 1) [1,3,4,5,6,7,8,9,10,11,12,13,14].

Fat globules enter the circulation and cause isolated pulmonary fat embolism or affect multiple organs. According to the mechanical theory, fat enters the venous circulation via intramedullary sinuses. An alternative explanation is the biochemical theory, in which fat breaks down due to lipase activation to free fatty acids and glycerol, initiating an inflammatory response and causing endothelial damage [15,16,17]. Fat embolism is the presence of fat in the lungs and peripheral microcirculation, with or without clinical consequences. Fat embolism syndrome (FES) is a potentially fatal manifestation of fat embolism with certain clinical and laboratory signs and symptoms, although the diagnosis is challenging [18,19,20]. The classical clinical triad of FES syndrome includes respiratory distress, neurological dysfunctions, and petechial rash [9]. The data on this condition’s mortality rate varies from 5.8%–10% to 30.2% [21,22,23]. Zenker described FES in 1861 in human patients [24]. There are different scoring and criteria systems: for example, Schonfeld’s scoring system, Gurd and Wilson’s criteria system, and Lindeque’s criteria (Table 2) [5,15].

The lungs, kidneys, and central nervous system are the most frequently involved organs, and pulmonary symptoms are the most common within 24 to 72 h after trauma.

C-reactive protein (CRP) is a well-known, atypical immunochemical marker associated with various pathological conditions, such as bacterial infections, sepsis, autoimmune disorders, malignancies, and myocardial infarction. It is produced primarily by hepatocytes through the regulation of Interleukin-6 (IL-6) and is a useful indicator in clinical practice [25]. CRP may be a diagnostic and prognostic index for cancer, an acute-phase protein that is not specific to cancer type. Malignancy can promote inflammatory responses, initiate tumor progression, and increase disease severity [26,27]. It seems that CRP and its isoforms have predictive value, not only in cases of de novo tumorigenesis but in also more advanced stages and metastases [28,29]. According to Hulman, CRP triggers calcium-dependent agglutination of very low-density lipoproteins (VLDL) and chylomicrons, causing non-traumatic fat embolism [30]. This study aimed to prove or exclude the presence of fat embolism and fat embolism syndrome in the deceased patients with malignancy and determine a possible role in the cause of death.

## 2. Materials and Methods

We saved tissue samples from cadavers who had been diagnosed with cancer and excluded all other possible origins of fat embolism or fat embolism syndrome, independently of gender. Thus, any diagnosis or condition listed in Table 1 was excluded, with the exception of bronchopneumonia and attempts at cardiopulmonary resuscitation [31]. Age was not a limiting factor during patient inclusion or exclusion. The sample collection period was September 2024 to March 2025. Over 600 autopsies were conducted, and 11 deceased individuals met the criteria for inclusion in this study. A total of 12 pieces of tissue samples were collected from each patient (1.5 × 1.5 × 1 cm maximum size): 5 pieces of the lungs (1 from each lobe), 4 pieces of the brain (1 from the right frontal lobe, 1 from corpus callosum, 1 from right basal ganglia, and 1 from left occipital lobe), 2 pieces of the kidneys (1 from each side middle section), and 1 piece of heart (left ventricle anterior wall). The autopsies took place 36–190 h after death. In five cases, frozen slides were performed immediately after autopsies, and in the other cases after 24–196 h. For the latter, the tissue samples were fixed on a cryostat chuck, dipped in petrol ether chilled by dry ice to −70 °C, and embedded with Epredia cryomatrix embedding resin (Fisher Scientific, Pittsburgh, PA, USA, product code: 12542716). These samples were stored in a freezer (−18 °C) until processing. Frozen sections of 5–8 μm were made using a cryotome (Thermo Scientific, Pittsburg, PA, USA). After 24 h of drying, the slides were immersed in 70% isopropanol. A quantity of 0.2–0.5 g solid Oil Red O (Sigma-Aldrich, Burlington, MA, USA, catalog number: O0625) was dissolved in 10 mL isopropanol (Molar Chemicals, Halásztelek, Hungary, 2-PROPANOL (IZOPROPANOL), catalog number: 00390-526-206), and 4 mL distilled water was added to 6 mL of the stock solution, which was then filtered before use. The Oil Red O solution was strained through filter paper and dispensed on the slides. After 10 min of staining with Oil Red O, the solution was drained off, and the sections were merged in 70% isopropanol and rinsed with distilled water for 1–2 min. After 1 min of staining with Mayer’s Alum Hematoxylin (Molar Chemicals, catalog number: 42514), tap water bluing was administered. The slides were then covered with a mounting medium. Two forensic experts/pathologists evaluated the slides separately under a light microscope (Leica, Wetzlar, Germany, 2500 DM).

Lung fat embolism was evaluated as per Falzi et al., modified by the Janssen score system (Table 3). 

Another set of tissue samples was also harvested from the above-mentioned organs, as well as from the suspected malignancies and possible metastases for usual Hematoxylin and Eosin (H&E) staining after a maximum of 5 days of formalin fixation (VWR Chemicals, Radnor, PA, USA, Formaldehyde 4% aqueous solution, buffered, catalog number: FOR010LAF59001).

We determined the details if we had no relevant or sufficient clinical or pathological data regarding the cancer type, grading, staging, and histological type (hematoxylin-eosin staining, immunohistochemical reactions) of the tumor. However, when we had data, we compared the autopsy results with clinical findings (Figure 1).

## 3. Results

Of 621 autopsies (including pathological and forensic autopsies), 15 seemed appropriate for sample collection, and only 11 matched our preset criteria (Table 4).

Nine of the specimens were male and two were female. The average and median ages were 66.4 years and 64 years, respectively. There were various types of cancer in different parts of the gastrointestinal tract (esophagus, stomach, colon, and rectum), prostate, liver, thymus, and lungs. The subepicardial fatty tissue was used as a positive internal control for Oil Red O lipid staining. In 5 out of 11 autopsies (45%), sporadic lung fat embolism was observed. In one of the five cases, there was sporadic pulmonary fat embolism in addition to the kidney (Figure 2). The latter case was one with no advanced-stage cancer.

Lung metastases were observed in three cases, liver metastases in two cases, and both lung and liver metastases in one case. In addition, osseous and pleural metastases were seen in one case each, and both metastases were seen in one case, while the adrenal gland and peritoneum were affected in one case each. The antemortem C-reactive protein level was observed in two cases. Cachexia was observed in all of the advanced stages of cancer. In one case, the detected malignancy was the third primary cancer.

There were only two cases with osseous metastases, and one of these two cases showed sporadic lung fat embolism. In our opinion, it is worth expanding the research from this point of view; however, in these cases, the mechanical background of the fat embolism, such as the interference of intramedullary expanding tumor, vasculature, and intramedullary fat, is also possible in addition to the biochemical form.

None of these patients had fat embolism as a lethal condition, and no fat embolism syndrome was observed.

In the four remaining cases (one female, three males, average and median age 62.5 years and 52 years, respectively), besides the cancerous primary disease, bacterial purulent bronchopneumonia was observed. Three patients had locally advanced transitional cell carcinoma of the bladder, well-differentiated squamous cell carcinoma of the tongue, and poorly differentiated squamous cell carcinoma of the esophagus, and one had advanced anaplastic cancer of the lung. Two of the four cases expressed punctiform, sporadic fat embolism of the lung, and in one case, in some fields of view, it was more disseminated and teardrop-like, but not in every field of vision at 25× magnification. In these cases, inflammatory disease could also increase CRP levels (Table 5).

## 4. Discussion

Although the mechanical theory of fat embolism formation, which leads to vessel occlusion and subsequent local organ damage, has been elucidated, the biochemical theory remains ambiguous. Trauma may result in elevated plasma lipase levels or, due to biochemical alterations, fat globules may enter the vasculature and decompose into free fatty acids and glycerol. This process potentially triggers a microvascular inflammatory response, which is a plausible hypothesis [1].

The increased presence and role of CRP in malignancies have been extensively studied. This immunochemical marker is one of the most important, and its potential role from diagnostic, biochemical, and pathophysiological points of view is incontestable [25]. CRP has different forms and can switch between various types, indicating evidence of various functions [32]. CRP ligand affinity is partly calcium-dependent, as in the case of pathogens, histones, glycans, etc., but without calcium, it can bind myelin essential proteins or poly-L-arginine [33,34]. In two of our autopsy cases, the antemortem CRP level was known within 24 h before death. These levels were significantly increased, at 149.6 mg/L and 103.37 mg/L. However, no evidence of fat embolism was found. Furthermore, the number of cases was insufficient for valid statistical analysis. The postmortem measurement of CRP levels may be useful. According to the literature, the pre- and postmortem levels of blood CRP depend on the analysis method (immunoturbidimetric or immunometric). The average reduction was 42–35%, and individually, the highest decrease was 74% [35]. CRP is a pre-diagnostic marker of lung cancer in current smokers and in small cell lung cancers. It correlates with size and staging, indicating a poor prognosis, whereas in adenocarcinomas, it indicates the paucity of treatment response with or without epidermal growth factor receptor (EGFR) in advanced stages [36,37,38]. In other solid tumors (e.g., esophageal, stomach, gynecological, pancreas, head and neck squamous cell carcinoma), the sometimes-notable CRP increase can indicate poor survival or the presence of metastases, determine treatment outcomes or tumor recurrence, and advanced stages, and correlates well with tumor size and disease progression. Preoperative CRP increase can also be linked to worse outcomes [25,26,39,40,41,42,43]. It also seems to be a promising immunohistochemical marker to differentiate between intrahepatic cholangiocarcinoma and other adenocarcinomas [42]. In addition, in leukaemias and lymphomas, it has been reported as a prognostic marker [44,45]. In 82% of our cases, when there was an advanced or locally advanced stage of cancer, we presumed an increased CRP level. However, the possible anergic state of end-stage patients can influence the immunological status and responses; therefore, we had to deal with the downregulation. This may be one of the answers to our study’s low number of pulmonary fat embolisms and the lack of fat embolism syndrome. Although the number of proper cases was low, we also raised the possibility that the type of primary cancer somehow regulates the appearance of fat embolism.

The strategies for the prevention of fat embolism in the case of a cancerous patient are limited or almost impossible, as the elevated CRP level is characteristic of the malignancies. However, the long-term application of corticosteroids increases the possibility of fat embolism, but when used as a prophylactic treatment, it seems effective in decreasing the free fatty acid level, stabilizing membranes. It can also inhibit the complement-mediated leukocyte aggregation in patients with long-bone fractures [15]. Steroids are widely used in oncology because of their anti-inflammatory, anti-swelling, and angiogenesis inhibition effects, both as a therapeutic treatment and for supportive care [46].

Fat embolism, particularly that leading to Fat Embolism Syndrome (FES), has significant implications for cancer patients due to several factors that can increase their risk and complicate their clinical course. While the clinical presentation poses a challenge for diagnosis through multi-organ dysfunctions, it also carries the potential for severe morbidity and mortality. Providing respiratory and hemodynamic support, along with close monitoring, is essential for accurate diagnosis. High awareness and prompt supportive care are critical for improving outcomes. Our findings indicate a need to increase the number of cases by extending the study period and including patients from diverse racial backgrounds, while also broadening the types of cancer studied and the ages of patients in our study group. By enhancing the understanding of distinct mechanisms of the disease, the unique physiological changes in cancer alter the course of presentation and severity of fat embolism. If certain types of cancer or treatment heighten the risk of fat embolism, this allows for better risk stratification and proactive measures. Interdisciplinary approaches can lead to improved preventive care and enhanced supportive care, while also mitigating inflammatory responses and promoting fat globule breakdown, which are critical in improving outcomes for cancer patients with fat embolism. In summary, this study enhances our understanding of fat embolism in cancer patients by exploring the unique aspects of this population.

### Limitations

The current study is limited by unintended bias during data collection, which stems from strict patient criteria, a consecutive limited number of cases, and time frame restrictions. These factors contribute to limitations in the female-to-male ratio, the spectrum of cancer types, and the representation of a younger population.

## Figures and Tables

**Figure 1 diseases-13-00174-f001:**
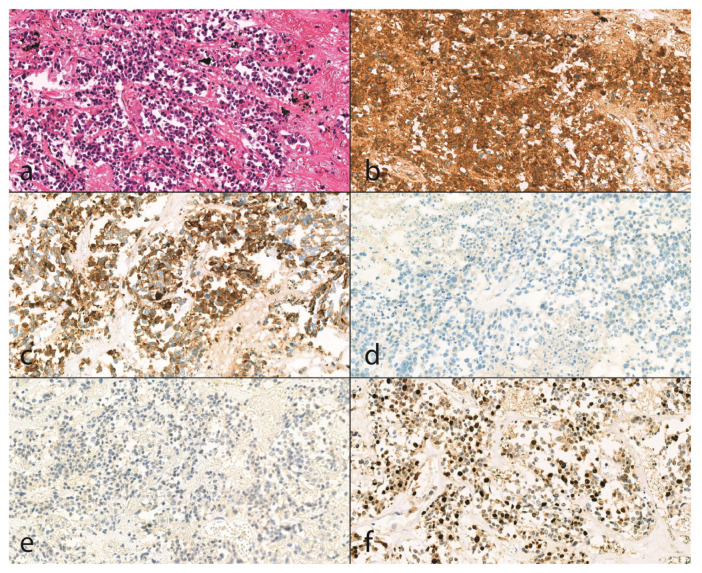
Histologic findings of an unknown malignancy case (20X magnification). (**a**): With hematoxylin and eosin staining (H&E), small cell neuroendocrine carcinoma is seen in biopsy material with an infiltrative growth pattern. The tumor consists of small cells with round to oval or elongated nuclei with a smooth nuclear membrane, finely dispersed chromatin, and no prominent nucleoli, and scant eosinophilic cytoplasm. The tumor cells form bundles or small clusters, and the tumor stroma is thin and fibrovascular. In some places, we can see individual cell necrosis. Positive immunoreactivity for (**b**) CD56 (Dako, CD56, clone: 123C3, catalog number: M7304, (1:200)) and (**c**) CK7 (Dako, Cytokeratin7, clone: OV-TL 12/31, catalog number: M7018, (1:1500)) with negative staining for (**d**) p63 (Dako, p63 protein, catalog number: M7317, (1:200)) and (**e**) WT1 (Leica Biosystems, Novoscastra Liquid Mouse Monoclonal Antibody Wilms’ tumor, product code: NCL-L-WT1-562, (1:25)) support the diagnosis of small cell carcinoma. Mitotic rate is high (**f**) Ki-67 (Dako, Ki-67 antigen, catalog number: M7240, (1:200)), proliferation index is 50%.

**Figure 2 diseases-13-00174-f002:**
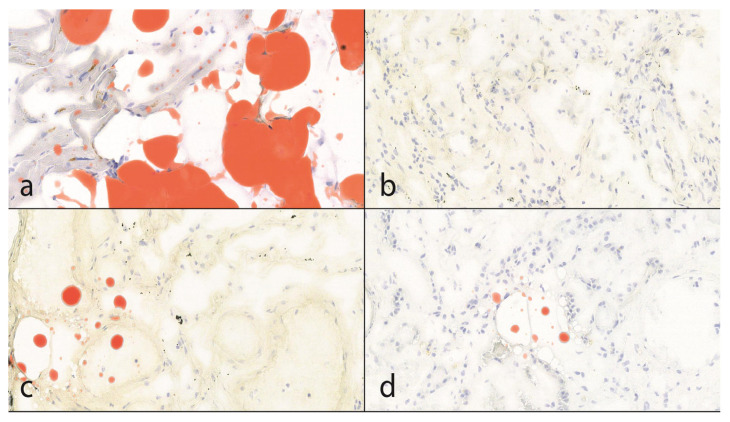
The results of lipid staining (25× magnification). (**a**)**:** The subepicardial fatty tissue is positive with Oil Red O lipid staining. This is also useful as an internal control. In the myocytes, lipofuscin formation can also be seen as evidence of former oxidative stress. (**b**,**c**): In 55% of the cases, there was no evidence of fat embolism; in 45%, sporadic, punctiform pulmonary fat embolism was detected. According to the Falzi et al. modified by the Janssen score system, the grade of the fat embolism was 0. (**d**)**:** In 1 case, punctiform, fat embolism of the kidney was observed.

**Table 1 diseases-13-00174-t001:** Possible causes of fat embolism.

Long bone and pelvis fractures [1,4,5,6]	Orthopedic surgeries [5,7]	Excessive soft tissue injuries [1,2,5]
Intra-osseus infusion [8]	Liposuction [1,2,4,5,6,7]	Burn injuries [1,2,5]
Bone marrow harvesting [5]	Sickle cell disease [1,2,4,5]	Acute osteomyelitis [1,4,5]
Bone marrow transplantation [2,5]	Pathologic bone fractures [10]	Epilepsy * [11]
Acute pancreatitis [1,2,4,5]	Diabetes mellitus [1,5]	Extracorporeal circulation [12]
Chronic pancreatitis [2,4,5]	Decompression sickness [1]	Intraosseous venography [13]
Fatty liver [2,4,5]	Carbon tetrachloride poisoning [1]	Bone marrow necrosis [14]
Lymphangiography [2]	Prolonged corticosteroid therapy [1,2,5]	Cardiopulmonary bypass [1]
Viral hepatitis [9]	Acute respiratory distress syndrome (ARDS) [4]	Panniculitis [5]
Median sternotomy [2]	Cyclosporin A solvent [5]	Heat stroke [1]

* Not directly related to epilepsy, but with the higher prevalence of bone fractures.

**Table 2 diseases-13-00174-t002:** Schonfeld’s criteria system for FES (for diagnosis scores more than 5).

Criteria/Symptoms	Score
Petechial rash	5
Diffuse alveolar infiltrates on chest X-ray	4
Hypoxaemia (PaO_2_ < 9.3 kPa)	3
Fever	1
Tachycardia (>120 bpm)	1
Tachypnea (>30 bpm)	1
Confusion	1

**Table 3 diseases-13-00174-t003:** Falzi et al. modified by the Janssen pulmonary fat embolism scoring system, grades, and criteria.

Shape of the Fat Globules	Compass of the Fat Embolism	Grade
Punctiform where relevant	No fat embolism, or sporadic	0
Teardrop-like	Every field of vision at 25× magnification	I
Sausage or lake-like	Disseminated form-no field of view without fat embolism	II
Antler-like	Massive, profuse form in every field of vision	III

**Table 4 diseases-13-00174-t004:** Synopsis of strictly selected deceased of the study, including age, gender, cause of death, autopsy findings, relevant clinical data, cancer type, grade, TNM (tumor, node, metastases) classification, cancer stage, and the result of Oil Red O lipid staining with the pulmonary fat embolism grade of the individuals.

Cases	Gender	Age (years)	Cancer type/Grade/TNM (tumor, node, metastases) classification/Morphological code/AJCC (American Joint Committee on Cancer) 8th or 9th edition (where applicable)/Stage/ICD 10 code	Result of Oil Red O staining/Fat embolism/Grade by Falzi modified by Janssen score system	Other relevant data/C-reactive protein level within 24 h before death (where available)
1.	male	63	Adenocarcinoma of the rectum (Invasive adenocarcinoma of the rectum with signet ring cells and ulcerated surface) Gr3 aT3aN1M1 M81403 (Stage IVC) C20	No fat embolism Grade 0	-cachexia -sacral decubitus ulcer -lung metastases -percutaneous nephrostomy implant, both sides -Sigmoid colostomy CRP: 149.6 mg/L
2.	female	67	Small cell carcinoma of the lung (SCLC) Gr3 aT4bN1M1 M80413 (Stage IV) C34.9	No fat embolism Grade 0	-cachexia -osseous, and lung metastases -pleural carcinosis -former breast carcinoma (1995-right, 2017-left, invasive ductal carcinoma of the breast) CRP: 103.37 mg/L
3.	male	56	Adenocarcinoma of the prostate Gr3 aT4N1M1 M81403 (Stage IV) C61	Sporadic fat embolism of the kidney Grade 0	-cachexia -osseous, lung, and liver metastases -percutaneous nephrostomy implant, both sides
4.	male	48	Dedifferentiated carcinoma of the large intestine (hepatic flexure) (Dedifferentiated partially mucinous invasive adenocarcinoma of the colon) Gr3 pT4N1cM1c M80203 (Stage IVC) C18.3	No fat embolism Grade 0	-cachexia -lung and adrenal gland metastases
5.	male	46	Adenocarcinoma of the lung Gr2 aT4N1M1c M81403 (Stage IVB) C34.9	No fat embolism Grade 0	-cachexia -pleural carcinosis -thoracic diaphragm metastases
6.	male	74	Adenocarcinoma of the stomach (Intestinal type invasive gastric adenocarcinoma) Gr3 aT4bN2M1 M81443 (Stage IV) C16.9	Sporadic fat embolism of the lung Grade 0	-cachexia -multiple liver metastases -pericardial effusion
7.	female	100	Renal cell carcinoma Gr1 aT1aN1M1 M83123 (Stage I) C64	Sporadic fat embolism of the lung and kidney Grade 0	-senium
8.	male	61	Adenocarcinoma of the stomach (Invasive gastric adenocarcinoma with signet ring cells) Gr3 aT4bN1M1 M81443 (Stage IV) C16.9	No fat embolism Grade 0	-cachexia -multiple liver and greater omentum metastases -peritoneal carcinosis
9.	male	64	Adenocarcinoma of the lung Gr2 aT1N0M0 M81443 (Stage IA) C34.9	No fat embolism Grade 0	-hydrothorax (two sided) -chronic fibrous pleurisy
10.	male	88	Undifferentiated (large cell) carcinoma of the thymus Gr3 aT3N0M0 M80203 (Stage IIIA) C37H0	Sporadic fat embolism of the lung Grade 0	-moderate atherosclerosis and coronary artery disease -sigmoid colon resection due to diverticulitis -preternatural anus
11.	male	67	Hepatocellular carcinoma of the liver Gr1 aT3N0M0 M81703 (Stage IIIA) C22	Sporadic fat embolism of the lung Grade 0	-moderate atherosclerosis and coronary artery disease -ischemic heart disease -former myocardial infarct -lung edema -nephrosclerosis -chronic bronchitis -chronic emphysema

**Table 5 diseases-13-00174-t005:** Summary of the deceased individuals who were excluded from the target group because of the coincidence of cancerous disease and purulent bronchopneumonia as a mutual background of CRP level increasing.

Cases	Gender	Age (years)	Cause of death Main autopsy findings	Relevant clinical data with ICD 10 (International Statistical Classification of Diseases and Related Health Problems 10th Revision)	Cancer type/Grade/TNM (tumor, node, metastases) classification/Stage/ICD 10 code	Oil Red O staining Results Pulmonary fat embolism grade
1.	female	85	right heart failure purulent bronchopneumonia, acute bronchitis, bladder cancer (transitional cell carcinoma), cachexia, nephrosclerosis, edema and atrophy of the brain, bilateral hydrothorax, severe general atherosclerosis, moderate coronary sclerosis, nephrosclerosis, myoma of the uterus, sacral and heel region decubitus ulcer	Bladder cancer, unspecified (C67.9)-status post TURBT (trans-urethral resection of bladder tumor) and BCG (Bacillus-Calmette-Guérin) application, Essential (primary) hypertonia (I10.H0), Hypothyreosis (E03.9), Rheumatoid arthritis (M06.9), In anamnesis: Right tibial shaft fracture (S82.2) in 2018, Right pertrochanteric fracture (S72.1) years before death	Transitional cell carcinoma of the urinary bladder-high grade (Gr3) aT3bN0M0 (Stage IIIA) C67.9	Sporadic fat embolism in the lung (punctiform) Grade 0
2.	male	61	multiorgan failure purulent bronchopneumonia, lung cancer, multiple metastases (with osseous metastases), pulmonary edema, cachexia	Essentiel (primary) hypertonia (I10.H0), Type 2 diabetes mellitus with unspecified complications (E11.8), Right eye cataract surgery (H26.90)	Anaplastic carcinoma of the right lung-high grade (Gr3) aT4aN3aM1c2 (Stage IV.B) C34.9	Sporadic fat embolism in the lung (punctiform) Grade 0
3.	male	52	right heart failure purulent bronchopneumonia, acute bronchitis, accessory spleen, mild general atherosclerosis, tongue cancer, cachexia, periprostatic venous thrombosis	ND (no data)	Well-differentiated squamous cell carcinoma of the tongue- low grade (Gr1) aT3N0M0 (Stage III) C02.2	No fat embolism Grade 0
4.	male	52	septic multiorgan failure purulent bronchopneumonia, esophagus cancer with lymph node metastases, cachexia	Gastroduodenitis, unspecified (K29.9), Diaphragmatic hernia without obstruction or gangrene (K44.9)	Poorly differentiated squamous cell carcinoma of the esophagus- high grade (Gr3) a T4bN2M0 (Stage IVA) C15.90	Punctiform, in some view sporadic and in some fields teardrop-like more disseminated fat embolism, but not in every field of view at 25× magnification Grade 0

## Data Availability

The authors confirm that the data supporting the findings of this study are available within the article.

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
