# Peer review of "The Relationship Between Non-Traumatic Fat Embolism and Fat Embolism Syndrome (FES) in Patients with Cancer"

_diseases, 2025, doi:10.3390/diseases13060174_

Round 1
Reviewer 1 Report
Comments and Suggestions for Authors
The research article is a study of the relationship between non-traumatic fat embolism and fat embolism syndrome in cancer patients. The study involves patients with various histological types and origins of cancer, as well as advanced cancer in 90% of the deceased patients. Tissue samples are taken to detect fat emboli. The results indicate that less than 50% of cases present with a punctate pulmonary fat embolism, which is not clinically significant, and furthermore no fat embolism syndrome has been identified. The authors' conclusions are fully supported by the results although the sample is very limited (in fact is it representative). Thus, the anergic state of patients at the end of life can influence the procedure. In the case of bone metastases, mechanical and biochemical factors may prevail and promote the formation of fat embolism.
Recommandations for publication;
- add references in Table 1
- The sample selection method must appear in part "2"
- Why not include younger people in the panel of deceased patients; under 50 years old.
- line 133 "The sample collection period was September 2024 to March 2025. Of 621 autopsies (including pathological and forensic autopsies), 15 seemed appropriate for sample collection, and only 11 matched our preset criteria (Table 4)." Why not extend the sample collection period to have a better male-female sample and to have younger people in the sample.
- Have you checked that in the sample selected, the people did not have other pathologies.
- Part 4 deserves more development.
- It is also important to clearly distinguish between a conclusion, which should conclude with perspectives.
Reviewer 2 Report
Comments and Suggestions for Authors
Dear Authors! Thank you for the opportunity to review your manuscript. Fat embolism and fat embolism syndrome is severe well-known complication of the traumatic patients and it's prevalence in other disease is not estimated.
The Authors provided the study to find fat embolism in oncology patients to confirm the known hypothesis of fat embolism development.
The Introduction explansvthe actuality and has the known reasons and diagnostic algorithm of this condition. The methods described well and illustrated by the figures as well as results.
The discussion is very small and should be extended and the comprehensive limitations section is required.
The Conclusion supports the study results
The references are relevant to the study
Round 2
Reviewer 1 Report
Comments and Suggestions for Authors
The authors having responded to my questions and suggestions, I therefore propose the publication of the article.